# Exploring the Utility of Autonomic Nervous System Evaluation for Stroke Prognosis

**Ilias Orgianelis** [1,†]**, Ermis Merkouris** [1,†]**, Sofia Kitmeridou** [1,†]**, Dimitrios Tsiptsios** [1,*]**, Stella Karatzetzou** [1]**, Anastasia Sousanidou** [1]**, Aimilios Gkantzios** [1]**, Foteini Christidi** [1]**, Efthymia Polatidou** [1]**, Anastasia Beliani** [1]**, Anna Tsiakiri** [1]**, Christos Kokkotis** [2]**, Stylianos Iliopoulos** [2]**, Konstantinos Anagnostopoulos** [2]**, Nikolaos Aggelousis** [2] **and Konstantinos Vadikolias** [1]

1   Neurology Department, Democritus University of Thrace, 68100 Alexandroupolis, Greece; iliaorgi@med.duth.gr (I.O.); ermimerk@med.duth.gr (E.M.); s.kitmer@gmail.com (S.K.); skaratzetzou@gmail.com (S.K.); anastasiasousanidou@gmail.com (A.S.); aimilios.gk@gmail.com (A.G.); christidi.f.a@gmail.com (F.C.); evthpola@med.duth.gr (E.P.); anasbeli1@med.duth.gr (A.B.); anniw_3@hotmail.com (A.T.); vadikosm@yahoo.com (K.V.)
2   Department of Physical Education and Sport Science, Democritus University of Thrace, 69100 Komotini, Greece; ckokkoti@affil.duth.gr (C.K.); iliopous@phyed.duth.gr (S.I.); kanagno@phyed.duth.gr (K.A.); nagelous@phyed.duth.gr (N.A.)
*   Correspondence: tsiptsios.dimitrios@yahoo.gr; Tel.: +30-694-432-0016
†   These authors contributed equally to this work.

**Abstract:** Stroke is a major cause of functional disability and is increasing in frequency. Therefore, stroke prognosis must be both accurate and timely. Among other biomarkers, heart rate variability (HRV) is investigated in terms of prognostic accuracy within stroke patients. The literature research of two databases (MEDLINE and Scopus) is performed to trace all relevant studies published within the last decade addressing the potential utility of HRV for stroke prognosis. Only the full-text articles published in English are included. In total, forty-five articles have been traced and are included in the present review. The prognostic value of biomarkers of autonomic dysfunction (AD) in terms of mortality, neurological deterioration, and functional outcome appears to be within the range of known clinical variables, highlighting their utility as prognostic tools. Moreover, they may provide additional information regarding poststroke infections, depression, and cardiac adverse events. AD biomarkers have demonstrated their utility not only in the setting of acute ischemic stroke but also in transient ischemic attack, intracerebral hemorrhage, and traumatic brain injury, thus representing a promising prognostic tool whose clinical application may greatly facilitate individualized stroke care.

**Keywords:** autonomic nervous system; autonomic dysfunction; heart rate variability; stroke prognosis; stroke outcome

## 1. Introduction

The term "stroke" refers to the sudden occurrence of a neurologic deficit, lasting longer than 24 h or resulting in death that can be linked to a specific vascular etiology [1,2]. Stroke not only constitutes the second most significant cause of mortality among adults, but it is also the leading cause of acquired disability, exhibiting a significant negative impact on the long-term functional independence of the survivors [3]. As two-thirds of stroke patients are older than 65 years, it is generally acknowledged that both stroke incidence and prevalence are constantly increasing within the worldwide aging population [4].

The need for rapid and accurate stroke prognosis is being highlighted by both the expected significant increase in stroke survivors' numbers and the detrimental effects of stroke on each patient's ability to independently perform activities of daily living (ADL). Estimating the chances of regaining a certain degree of motor function early after a stroke is considered to be of great benefit when considering the type and duration of treatment

techniques. In addition, accurate prognostication may facilitate the development of reasonable goals, properly allocate time and resources, as well as lead to an individualized recovery plan [5].

Numerous clinical assessment techniques have been implemented as prognostic indicators aiming at providing suitable information and enhancing the personalization of stroke care in terms of long-term outcome prediction post-stroke [6,7], ABCD2 score (age, blood pressure, clinical features of transient ischemic attack, duration of symptoms, and presence of type-2 diabetes) [8] and National Institutes of Health Stroke Scale (NIHSS) [9] being among them. Similarly, several neurophysiological techniques have been utilized to provide useful information about each patient's recovery capacity [10]. Nevertheless, given the diversity of stroke clinical manifestations, relying solely on clinical data has not proven to be sufficient for precise stroke prognostication [11]. Hence, to accurately forecast each stroke survivor's recovery potential, biomarker-based methods in a setting of an acute stroke may therefore be used as an adjunctive prognostic tool [12–14]. The term "biomarker" is defined by the Biomarkers Definitions Working Group as "a characteristic that is objectively measured and evaluated as an indication of normal biological processes, pathogenic processes, or pharmacologic responses to a therapeutic intervention" [15]. The optimal prognostic tool should have high specificity and sensitivity in identifying individuals with recovery potential to ensure that these patients receive proper rehabilitation. It is of great importance that the utility of biomarkers is not restricted to the acute and subacute phase of phases, as a prediction of recovery using suitable biomarkers may also be helpful even in the chronic phase [16].

Until recently, various biomarkers have been investigated in terms of prognostic accuracy poststroke, heart rate variability (HRV) being among them. HRV refers to beat-by-beat heart variations as a result of a physiological variation in the duration of the R-R intervals between sinus beats [17,18]. In 1965 Hon and Lee introduced HRV as a method aiming at examining autonomic function by correlating fetal discomfort with variations in heart rate [17]. HRV is an easily obtainable non-invasive method of studying autonomic function, simply requiring an electrocardiogram (ECG) [2,17,18]. More specifically, HRV quantifies the sympathetic-vagus tone at the sino-atrial level and is, therefore, a commonly utilized biomarker in an attempt to evaluate the function of the autonomic nervous system (ANS) [18,19]. The HR is a result of the autonomic tone predominance and is influenced primarily by parasympathetic and secondarily by sympathetic fibers, as well as humoral factors (renin-angiotensin system, adrenomedullary catecholamines, hormones, direct neural innervations of the heart) [7,20]. HRV depends on age, gender, and ethnicity [17]. The 24-h Holter recordings in a group of 20 to 70-year-olds showed an age-related decline in every parameter [21]. Furthermore, healthy women's HRV is significantly lower than healthy men's, and young African Americans have been found to have lower parasympathetic activity [21].

Several methods used to analyze HRV are based on time domain analysis, frequency domain analysis, and nonlinear methods of analysis [2]. Time domain techniques allow measurement of either the immediate heart rate or the intervals between successive normal QRS complexes (normal-to-normal R-R interval, NN) in a continuous ECG recording. [8]. More useful in a clinical setting parameters of HRV, using statistical operations to evaluate R-R intervals, are mean heart rate, the standard deviation of normal R-R intervals (SDNN), the standard deviation of the average normal R-R intervals (SDANN), the root mean square of successive R-R interval differences (rMSSD) and the percentage of normal R-R intervals that differ by 50 ms (pNN50) [2,8,17]. Whereas rMSSD is correlated with vagal-mediated regulation, SDNN reflects the overall HRV [2].

Frequency domain analysis is based on spectral analysis of fluctuation of autonomic tone. The spectral analysis breaks down a signal into its frequencies [2,8]. High frequency (HF), from 0.15 to 0.4 Hz, represents the parasympathetic tone and is linked to the respiratory cycle, low frequency (LF), from 0.04 to 0.15 Hz, reflects baroreceptor activity (afferent) and is considered to represent mostly the sympathetic, but also the parasympathetic tone

(efferent). The role of very low frequency (VLF) is still unknown but is considered to reflect the integrative effect of various controllers, such as vagal to humoral effects [2,8,17,19]. There is also the LF/HF ratio, which indicates the index of interaction between sympathetic and vagal activity. The importance of nonlinear measurements as a prognostic tool is still unclear [17].

Age and the severity of the first stroke are currently thought to be the two most significant indicators of functional improvement and eventual home discharge in ischemic stroke patients. However, after the acute phase of ischemic stroke, both decreased HRV and impaired cardiac heart baroreceptor sensitivity (BRS) have been linked to poor clinical outcomes [22–24].

The cardiovascular autonomic regulatory system appears to sustain considerable, long-lasting damage from ischemic stroke, which manifests as abnormalities in HRV [23–26]. A common neural circuit simultaneously modifies the somatic-motor and autonomic nervous systems. This circuit is located in gray matter, delivering polysynaptic projections to the relevant somatic motor and autonomic nerve regions. The vagus nerve is regulated either directly or indirectly by the sensorimotor cortex and corticospinal pathways, which also initiate and coordinate upper and lower extremity movements [23,27–29]. Damage to the sensorimotor cortex and corticospinal cord pathway after stroke may impair vagus nerve function and lower heart rate. Hence, HRV is a proximal biomarker of the integrity of cortical pathways associated with motor impairments of the affected upper and lower limbs, and stroke survivors with high HRV generally require less assistance in performing ADL [23,27–30].

In stroke patients, cerebral autoregulation is often disturbed because blood pressure changes due to AD can directly affect cerebral blood flow (CBF). Hence, early mobilization could induce hypotension followed by a decrease in local CBF in brain tissue with impaired perfusion [31,32]. In addition, an increase in BP can disrupt the blood–brain barrier, leading to cerebral microbleeds and edema [31]. These abnormalities may negate the benefits of neurovascular repair and prevention of disuse syndrome and lead to neurological deterioration [31,32].

The poor stroke prognosis has been associated with a progressive loss of general autonomic regulation, a decrease in parasympathetic tone, and a progressive shift toward sympathetic dominance [12,33,34]. Sympathetic overactivity and increased norepinephrine can lead to stroke-induced immune suppression, which predisposes patients to infection [35,36]. According to experimental and clinical data, sympathetic hyperactivity is related to the production of pro-inflammatory cytokines, decreased lymphocyte activation, a switch from Th1 to Th2 cytokine predominance, and immunosuppressive syndrome related to stroke. Sympathetic hyperactivity and decreased BRS with subsequent immunosuppression have been associated with the occurrence of poststroke infections and linked to secondary brain injury [33,35,36]. In contrast to sympathetic overactivity, the parasympathetic peripheral cholinergic pathway is activated. Non-neuronal cells, including lymphocytes, are an important source of ACh that can directly suppress macrophages and inhibit their production of pro-inflammatory cytokines by binding $\alpha$7 nicotinic acetylcholine receptors ($\alpha$7nAChR), thus downregulating TNF-$\alpha$ produced by activated macrophages. However, the production of anti-inflammatory cytokines is maintained [35,37], leading to the "Central nervous system injury-induced immune deficiency syndrome" [37].

Both cardiac interoception and the efferent cardiovascular response to emotional experience are mediated by the insular cortex, which is an integral part of the central ANS. Sympathetic overactivity and decreased parasympathetic modulation as causes of cardiac dysfunction are among the most commonly discussed mediators and can be observed even when the insular cortex itself is not affected [38]. Due to the change in autonomic tone, blood pressure and heart rate are impaired, leading to increased susceptibility to cardiac side effects [14,15,38–40].

There is much evidence supporting that autonomic dysfunction leads to myocardial infarction (MI) and heart failure (HF) [38,41,42]. Firstly, patients with high activity in the

amygdala developed stress cardiomyopathy earlier than those with lower activity [38]. Furthermore, exercise and other conditions associated with sympathoexcitation are important triggers for MI. It is therefore assumed that overactivity of the sympathetic nervous system is closely associated with the disease [40]. In preclinical studies, increased circulating catecholamines are associated with the loss of noradrenergic nerve terminals in failing ventricles, leading to HF [38,39]. In addition, increased release and suppressed reuptake result in high extracellular norepinephrine (NE), which is detrimental to the heart. Acetylcholine (Ach) replaces NE in some sympathetic neurons as the disease progresses, and neuropeptide Y production and release increase, possibly leading to pathology. Remodeling of axon branches in the heart, including degeneration and regrowth, as well as larger cell bodies and dendritic arbors inside stellate and intracardiac ganglia, are morphological changes that add to neurochemical plasticity. Electrical remodeling includes increased stellate ganglia excitability, different firing rates for sensory and vagal afferents, and connections within the intracardiac ganglia. These changes affect the heart, causing downregulation of β1- and upregulation of β2-adrenoreceptors and alteration of cardiac electrophysiology [39]. In both conditions, beta-adrenergic blockers are the most established autonomic intervention associated with improved outcomes, indicating the role of ANS [38,40].

AD may imbalance autonomic responses at the cardiac level and lead to an increased risk of arrhythmia, with atrial fibrillation (AF) being the most common [41]. Both sympathetic and parasympathetic nerves are thought to be involved in the creation of AF. Studies have shown a shift toward a lower density of cholinergic nerves and a higher density of adrenergic nerves in chronic AF. There was also evidence of sympathetic hyperinnervation in patients with chronic AF. Before the onset of AF, nerve recordings from the stellate ganglia revealed enhanced sympathetic and vagal nerve output. [40]. Thus, the sympathetic dominance caused by AD could be the substrate for AF [13,24,33,34,40].

HRV has been thought of as a physiological indicator for emotional regulation and psychological well-being [24] and has also been associated with depressive disorder. Post-stroke depression (PSD) is common among stroke survivors and can affect functional recovery and quality of life [42,43]. HRV was reduced in depressed versus nondepressed patients, and higher sympathovagal balance (LF/HF) was linked to higher severity of the depressive symptoms, providing evidence that PSD is connected to lower parasympathetic activation [43]. The ability of depressed patients to adapt their ANS activity may be limited to challenging environments, and they fail to activate adaptive resources [42]. This can lead to phenotypic depressive symptoms such as exhaustion, fatigue, stress reactivity, and disturbed sleep patterns [42,44] (Figure 1).

Considering the urgent need for accurate prognosis provided early after stroke and the potential role of HRV in assessing the recovery potential of each individual, the objective of this study was to review all the relevant literature published within the last decade referring to HRV as a prognostic tool after stroke.

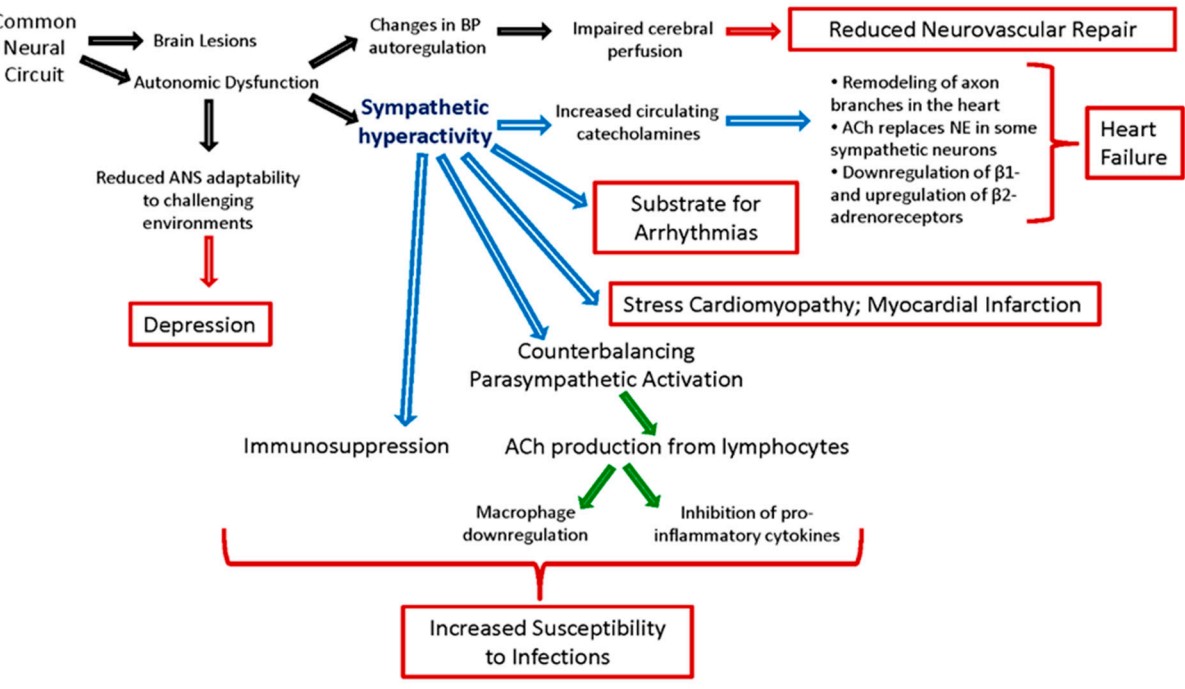

**Figure 1.** Pathophysiological pathways linking autonomic dysfunction to stroke outcome.

## 2. Materials and Methods

The Preferred Reporting Items for Systematic Reviews and Meta-Analyses (PRISMA) checklist [CRD42023417275] was used to guide this study. Our study's methods were a priori designed.

### 2.1. Search Strategy

Two databases (MEDLINE and Scopus) were selected to carry out the present literature search, which was conducted by one investigator (IO). To trace all relevant studies published between 1 January 2012 and 31 December 2022, the following keywords were used: "autonomic dysfunction" OR "autonomic dysfunction" OR "pupillometry" OR "heart rate variability" OR "baroreceptor sensitivity" OR "Ewing" OR "sympathetic" OR "parasympathetic" OR "orthostatic hypotension" AND "stroke prognosis" OR "stroke recovery" OR "stroke outcome" OR "stroke prediction". All retrieved articles were also hand searched for any further potential eligible articles. Any disagreement regarding the screening, or selection process, was solved by a second investigator (EM) until a consensus was reached.

### 2.2. Selection Criteria

Only full-text original articles published in the English language were included. Secondary analyses, reviews, guidelines, notes, errata, letters, meeting summaries, comments, unpublished abstracts, or studies conducted on animals were excluded. There was no restriction on study design or sample characteristics.

### 2.3. Data Extraction

Data extraction was performed using a predefined data form created in Excel. We recorded the author, year of publication, biomarker, type of stroke, number/ mean age of participants, medication, comorbidities/risk factors, previous stroke, follow-up time, time of assessment, the scale of stroke severity and prognosis, and finally, the main results of each study.

*2.4. Data Analysis*

No statistical analysis or meta-analysis was performed due to the high heterogeneity among the studies. Thus, the data were only descriptively analyzed.

**3. Results**

*3.1. Database Searches*

Overall, 3526 records were retrieved from the database search. Duplicates and irrelevant studies were excluded; hence, a total of 2777 articles were selected. After screening the full texts of the articles, 45 studies were eligible for inclusion (Figure 2).

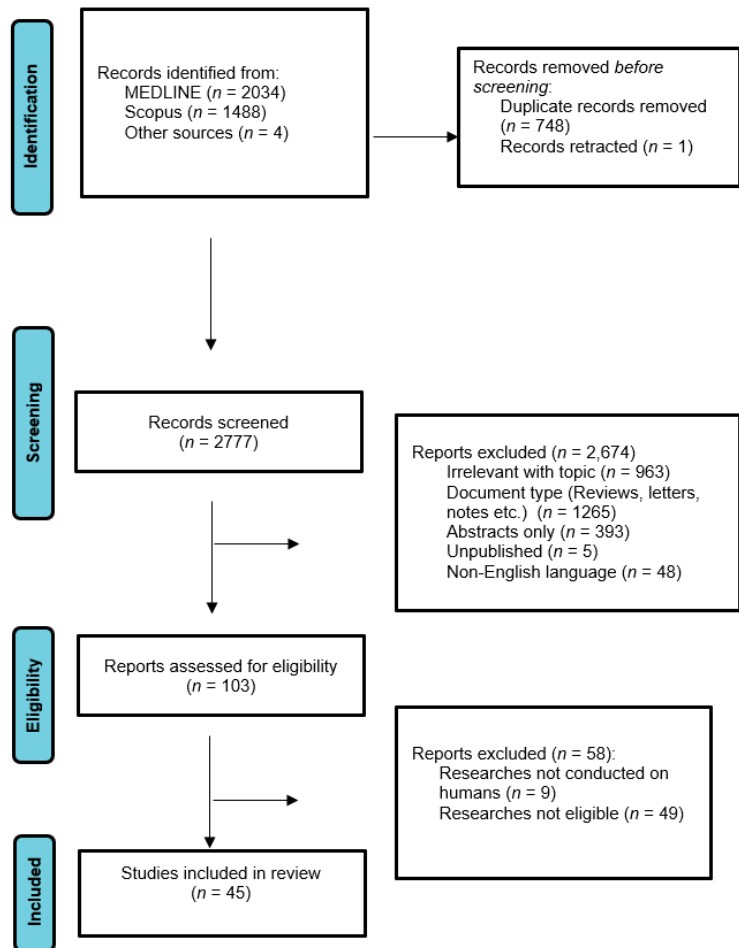

**Figure 2.** Study flow diagram (PRISMA flowchart).

*3.2. Study Characteristics*

Forty-five publications fulfilled our inclusion criteria. Thirty-two studies focused entirely on ischemic stroke (IS), five included patients with either IS or hemorrhagic stroke (HS), three had only patients with intracerebral brain hemorrhage, one study reported only on IS or HS or lacunar stroke (LS) or unknown origin (UO) stroke patients, one enrolled only patients with IS or HS or US, one enrolled patients with IS or transient ischemic attack (TIA) and one included patients with acquired brain injury. Concerning the origins of the studies, twenty-six were from Asia, nine came from Europe, six were from America, and three were from more than one continent (Table 1).

**Table 1.** Characteristics of the 45 included studies.

| Authors, Year of Publication | Type of Study | Biomarker | Type of Stroke | Number of Participants/Mean Age (y)/Gender (M/F) | Medication (*n*) | Comorbidities/ Risk Factors (*n*) | Previous Stroke | Follow-Up Time | Time of Assessment | Scale of Stroke Severity and Progno-sis/Clinical Outcome | Main Results |
|---|---|---|---|---|---|---|---|---|---|---|---|
| | | | | | | **Early Neurological Deterioration (within 1st week)** | | | | | |
| Shimada et al., 2022 [31] | Longitudinal | BP, HR | IS | 135/72.6 ± 12.5 95M/40F | - | Hypertension (15) Atrial fibrillation (15) Dyslipidemia (32) Diabetes mellitus (32) | NM | Until discharge | On admission | NIHSS (at onset) Barthel index | The fluctuation of the vital index up until the moment of transfer may be used to predict neurological deterioration following transfer to a wheelchair, which may result in a lengthy hospital stay. |
| Han et al., 2020 [45] | Longitudinal | HRV | AIS | 3447/68.6 ± 12.9 1991M/1.456F | Antihypertensive therapy (2014) Antiplatelet therapy (252) Anticoagulation therapy (37) Antiglycemic therapy (649) Statin therapy (104) Thrombolysis treatment (85) | Hypertension (2700) Diabetes mellitus (892) Coronary artery disease (193) Atrial fibrillation (532) | 772 | Until discharge | <7 days from onset | NIHSS (at baseline) | In AIS patients without AF, higher RHR at admission was independently related to in-hospital mortality. |
| Vistisen et al., 2014 [46] | Longitudinal | HRV | ABI | Patients; 49/52.7 ± (15.9 27M/22F Control; 49/52.6 ± 13.7 25M/24F | Calcium antagonists (14) Beta-blockers (6) ACE inhibitors (4) Antihypertensive drug (3) Amiodarone (3) | Type 1 diabetes (2) Type 2 diabetes (1) | - | Until discharge | 6 pm and 10 pm on the admission day, as well as 2 am and 6, am the following day | Functional Independence Measure (FIM) Rancho Los Amigos Scale (RLAS) | In a sample of heterogenic ABI patients admitted for neurorehabilita-tion, HRV was significantly decreased. |

| Authors, Year of Publication | Type of Study | Biomarker | Type of Stroke | Number of Participants/Mean Age (y)/Gender (M/F) | Medication (*n*) | Comorbidities/ Risk Factors (*n*) | Previous Stroke | Follow-Up Time | Time of Assessment | Scale of Stroke Severity and Prognosis/Clinical Outcome | Main Results |
|---|---|---|---|---|---|---|---|---|---|---|---|
| Chen et al., 2011 [47] | Longitudinal | HRV | IS (LAA and LAC) | Patients; 50/64.2 ± 11.8 27M/23F Control; 19/66.1 ± 8.5 7M/12F | Calcium channel blockers (P. 7/C. 2) ACEI or ARB (P. 3/C. 3) Other BP-lowering medication (P. 5/C. 0) | Smoking (P. 18/C. 3) Diabetes (P. 16/C. 7) Hypertension (P. 37/C. 15) Hyperlipidemia (P. 38/C. 14) | - | 7 days after admission | Within 48 h after stroke onset | NIHSS (at baseline and on the 7th day) | Patients in the LAA group showed lower parasympathetic activity than those in the LAC group, although they had more sympathetic activity. Depressed parasympathetic activity was linked to a higher probability of poorer early outcomes in the LAA group. |
| Huang et al., 2023 [48] | Cross-sectional | BPRO | AIS | 211/62.8 ± 10.8 136M/75F | CCB (114) ACEI (5) ARB (27) α-blockers (7) Beta-blockers (8) Diuretics (10) | Smoking (73) Alcohol (39) Hypertension (137) Diabetes (55) Coronary artery disease (16) Atrial fibrillation (5) Hyperlipemia (103) Hyperhomocys-teinemia (66) | 27 | 5 days | On admission | NIHSS | In hospitalized AIS patients, the repeatability of OH is low, and the simultaneous occurrence of OH and OHTN is frequent. |
| Usui H. and Nishida Y. et al., 2015 [49] | Longitudinal | HRV | IS | 12/67.7 ± 14.8 7M/5F | - | - | NM | Until discharge | 2–4 months after stroke | - | The VLF component of HRV and PA were found to be positively correlated in stroke patients. |

**Table 1.** *Cont.*

| Authors, Year of Publication | Type of Study | Biomarker | Type of Stroke | Number of Participants/Mean Age (y)/Gender (M/F) | Medication (*n*) | Comorbidities/ Risk Factors (*n*) | Previous Stroke | Follow-Up Time | Time of Assessment | Scale of Stroke Severity and Progno-sis/Clinical Outcome | Main Results |
|---|---|---|---|---|---|---|---|---|---|---|---|
| Park et al., 2022 [50] | Cross-sectional | HRV | IS, HS | 426/67.03 ± 13.03 197M/229F | Beta-blockers (87) Calcium channel blocker (193) ACEI/ARB (199) Diuretics (228) | Hypertension (260) Arrhythmia (74) Diabetes mellitus (158) Dyslipidemia (325) Coronary artery disease (22) Heart failure (70) | - | Until discharge | Within three months of onset | Mini-Mental State Examination | Blood levels of transferrin, prealbumin, and albumin were linked to ANS function as determined by HRV, and their deficit may be a determinant of how severe ANS dysfunction is in stroke patients. |
| Heinz et al., 2020 [51] | Cross-sectional | HRV, VSBP | IS, HS | 12/59 ± 7.00 8M/4F | Beta-blockers (6) Diuretics (2) Calcium channel blockers (3) ECA inhibitors (2) ARB (6) Diabetes T2 medication (3) Medication for cholesterol (3) Medication for arthrosis (1) Medication for spasticity (1) | Diabetes (3) Hypertension (10) Positive HIV (1) High cholesterol levels (3) | NM | Until discharge | First day of treatment | Fugl-Meyer lower limb scale | Except for intragroup comparison, which shows increased engagement in parasympathetic modulation in the group receiving active TtDCS, the treatment has no immediate impact on HRV and VSBP. |

**Table 1.** *Cont.*

| Authors, Year of Publication | Type of Study | Biomarker | Type of Stroke | Number of Participants/Mean Age (y)/Gender (M/F) | Medication (*n*) | Comorbidities/Risk Factors (*n*) | Previous Stroke | Follow-Up Time | Time of Assessment | Scale of Stroke Severity and Prognosis/Clinical Outcome | Main Results |
|---|---|---|---|---|---|---|---|---|---|---|---|
| Huang et al., 2017 [52] | Cross-sectional | HRV | IS, HS | Patients; 30/65.3 ± 8.4 12M/18F Control; 152/60.4 ± 11.7 69M/83F | Beta–blockers (P. 7/C. 25) Calcium channel blockers (P. 6/C. 30) ACEI or ARB (P. 4/C. 26) | Smoking (P. 3/C. 5) Diabetes mellitus (P. 26/C. 58) Hypertension (P. 25/C. 89) Coronary artery disease (P. 22/C. 28) | 30 | Until discharge | Before hemodialysis | - | During hemodialysis, there may be suppressed autonomic nervous reactions against volume unloading, which may worsen outcomes for hemodialysis patients and those who have had a stroke in the past. |
| Xiong et al., 2017 [53] | Longitudinal | HRV | IS | Patients; 48/64.69 ± 11.26 43M/5F Control; 14/59.36 ± 2.24 6M/8F | - | Hypertension (P. 10/C. 0) Diabetes (P. 21/C. 0) Hyperlipidemia (P. 29/C. 0) Smoking (P. 27/C. 0) Alcohol (P. 8/C. 0) Ischemic heart disease (P. 3/C. 0) Large artery disease (P. 39/C. NA) Small vessel disease (P. 9/C. NA) | 13 | Until discharge | Within 14 days of stroke onset | NIHSS (on admission) | Patients displayed an elevated beat-to-beat HRV following ECP, irrespective of the side of the ischemia. Moreover, after ECP in patients with right-sided subacute stroke, sympathetic and parasympathetic cardiac modulations increased. |
| Xu et al., 2016 [54] | Longitudinal | HR | AHIS | Patients: 63/71 ± 12 38M/25F Control: 50/68 ± 11 24M/25F | - | Hyperlipidemia (C.16/P. 18) Diabetes mellitus (C. 12/P. 21) Hypertension (C. 35/P. 50) Smoking (P. 27/C. 20) Alcohol (P. 4/C. 5) | - | Until discharge | Within 72 h after stroke | NIHSS (72 h after stroke) | In individuals with hemispheric infarction, both the DC and AC heart rate reduced, showing a reduction in both vagal and sympathetic regulation. The severity of the stroke was correlated with both DC and AC. |

**Table 1.** *Cont.*

| Authors, Year of Publication | Type of Study | Biomarker | Type of Stroke | Number of Participants/Mean Age (y)/Gender (M/F) | Medication (*n*) | Comorbidities/ Risk Factors (*n*) | Previous Stroke | Follow-Up Time | Time of Assessment | Scale of Stroke Severity and Prognosis/Clinical Outcome | Main Results |
|---|---|---|---|---|---|---|---|---|---|---|---|
| | | | | **Early Stroke Outcome (<1 Month)** | | | | | | | |
| Chidambaram et al., 2017 [55] | Longitudinal | HRV | IS/HS | 97/60.84 ± 14.12 56M/41F | - | - | - | 30 days | On admission | NIHSS (on admission and on the 30th day) | Patients with autonomic dysfunction also had higher blood pressure readings and increased morbidity and death during the acute phase of stroke. |
| Tsai et al., 2019 [28] | Longitudinal | HRV | AIS | Patients; 34/63.2 ± 8.7 26M/12F Control; 18/59.6 ± 7.8 11M/7F | - | Hypertension (26) Diabetes mellitus (19) Dyslipidemia (23) Coronary artery diseases (2) | NM | 1 month | Upon enrollment | NIHSS (on admission) | Patients who have brainstem or big hemisphere infarction show greater blunting BRS than those who have lacunar infarction, which might help identify patients who may be at risk for poor outcomes. |
| Lai et al., 2015 [56] | Longitudinal | BRS | ICH | Patients; 35/59.60 ± 13.47 25M/10F Control; 30/59.87 ± 6.52 20M/10F | - | Hypertension (26) Diabetes mellitus (7) Coronary artery disease (2) Hyperlipidemia (3) Smoking (6) Alcohol (4) Coagulopathy (1) | - | 30 days | Upon enrollment | mRS (at 30 days) | BRS value at admission is a more accurate indicator of outcome than the admission Glasgow coma score. |

**Table 1.** *Cont.*

| Authors, Year of Publication | Type of Study | Biomarker | Type of Stroke | Number of Participants/Mean Age (y)/Gender (M/F) | Medication (*n*) | Comorbidities/ Risk Factors (*n*) | Previous Stroke | Follow-Up Time | Time of Assessment | Scale of Stroke Severity and Prognosis/Clinical Outcome | Main Results |
|---|---|---|---|---|---|---|---|---|---|---|---|
| | | | | | | **Short-Term Stroke Outcome (<3 Months)** | | | | | |
| Sethi et al., 2015 [23] | Longitudinal | HRV | IS, HS | 13/61 ± 12 7M/6F | Beta-blockers (3) Calcium channel blocker (4) | Hypertension (6) Kidney stones (1) Pneumonia (1) Stomach ulcers (1) Type 2 diabetes mellitus (3) Hypercholesterolemia (1) Tubal ligation (1) Hyperlipidemia (2) Hyperthyroidism (1) Hypothyroidism (2) Secundum atrial septal defect (1) Four-vessel bypass (1) Appendectomy (1) Hysterectomy (1) Rheumatoid arthritis (1) | - | 3 months | Upon admission | NIHSS (at baseline) | A promising measure to investigate the mechanisms underlying motor recovery after stroke is HRV, which has a good correlation with motor outcome after stroke. |

**Table 1.** *Cont.*

| Authors, Year of Publication | Type of Study | Biomarker | Type of Stroke | Number of Participants/Mean Age (y)/Gender (M/F) | Medication (*n*) | Comorbidities/ Risk Factors (*n*) | Previous Stroke | Follow-Up Time | Time of Assessment | Scale of Stroke Severity and Prognosis/Clinical Outcome | Main Results |
|---|---|---|---|---|---|---|---|---|---|---|---|
| Zhang et al., 2019 [57] | Longitudinal | HRV | IS, HS | 64/59.05 ± 10.76 45M/19F | - | Hypertension (51) Diabetes (31) Coronary heart disease (15) Hyperlipidemia (45) Smoking (33) | - | 3 months | 1–3 months after stroke onset | - | There was no association between HRV values and the improvement in activities of daily living for stroke patients during the chronic rehabilitation phase, although there was a strong correlation between HRV parameters and the restoration of motor function. |
| Graff et al., 2013 [58] | Longitudinal | HRV, BP, RR | IS | 63/62 (30–87) 44M/19F | - | - | NM | 90 days | Within first 7 days after the onset of stroke symptoms | NIHSS (at baseline and on the 7th day) mRS (on the 7th and 90th day) | In the acute period of ischemic stroke, HRV measurements, but not blood pressure variability, distinguish groups with varied neurological outcomes. Second, in the immediate phase of ischemic stroke, a quicker respiratory rate is linked to a worse functional prognosis. |
| Szabo et al., 2018 [59] | Longitudinal | HRV | ICH | 47/60.8 ± 16.5 27M/20F | Beta-blockers (7) ACEI or AT (15) Calcium antagonists (6) | Hypertension (35) Amyloid angiopathy (4) Arteriovenous malformation (5) Coagulopathy (3) | - | 3 months | Within 24 h after stroke onset | NIHSS (on admission and at 10 days) mRS (at 3 months) | Autonomic changes appear to be present in acute ICH and are independently linked to poor outcome. |

**Table 1.** *Cont.*

| Authors, Year of Publication | Type of Study | Biomarker | Type of Stroke | Number of Participants/Mean Age (y)/Gender (M/F) | Medication (*n*) | Comorbidities/ Risk Factors (*n*) | Previous Stroke | Follow-Up Time | Time of Assessment | Scale of Stroke Severity and Prognosis/Clinical Outcome | Main Results |
|---|---|---|---|---|---|---|---|---|---|---|---|
| Tobaldini et al., 2019 [60] | Longitudinal | HRV | AIS | 41/68.0 ± 12.8 28M/13F | Beta-blockers (12) | Hypercholesterolemia (21) Hypertension (28) Diabetes mellitus (9) Heart failure (3) History of atrial fibrillation or flutter (3) Smoking (12) | 15 | 3 months | At the time of presentation in the emergency department | NIHSS (at the onset) mRS (3 months after stroke onset) | In the very initial stages of AIS, a loss of sympathetic oscillation may be reflected by a decreased 0V% and an elevated 2UV%, which may indicate a worse 3-month outcome. |
| Miwa et al., 2018 [61] | Longitudinal | HRV | ICH | 994/62 615M/379F | Antihypertensives (490) | Hypertension (788) Atrial fibrillation (35) Ischemic heart disease (43) Dyslipidemia (240) | 164 | 3 months | During the initial 24 h post-randomization | NIHSS (at baseline) mRS (at 90 days after randomization) | Throughout the first 24 h, increased mean HR and HR-ARV were independently linked to a poor prognosis in acute ICH. At 24 h, HR-ARV was linked to hematoma growth. |
| Jeong et al., 2016 [62] | Longitudinal | HR | AIS | 246/67.4 ± 12.8 132M/114F | Thrombolytic treatment (21) | Hypertension (179) Diabetes mellitus (74) Atrial fibrillation (57) Hyperlipidemia (78) Smoking (72) | 72 | 3 months | 7 days after onset | NIHSS (on admission) mRS (3 months after stroke) | Individuals who experience more tachycardia throughout their stay in the stroke unit have poorer functional outcomes. |
| Xiong et al., 2018 [63] | Longitudinal | Autonomic function (Ewing Battery) | IS | 150/66.4 ± 9.9 106M/44F | Beta-blockers (26) Calcium-channel blocker (79) Antiplatelet (132) HMG-CoA reductase inhibitor (124) | Hypertension (93) Diabetes mellitus (51) Ischemic heart disease (15) Hyperlipidemia (65) Smoking (64) Alcohol (34) | 34 | 3 months | Within 7 days of stroke symptom onset | NIHSS (on admission) mRS (3 months after stroke onset) Barthel index | Poor functional prognosis following acute ischemic stroke is predicted by autonomic dysfunction as measured by the Ewing battery. |

**Table 1.** *Cont.*

| Authors, Year of Publication | Type of Study | Biomarker | Type of Stroke | Number of Participants/Mean Age (y)/Gender (M/F) | Medication (*n*) | Comorbidities/ Risk Factors (*n*) | Previous Stroke | Follow-Up Time | Time of Assessment | Scale of Stroke Severity and Prognosis/Clinical Outcome | Main Results |
|---|---|---|---|---|---|---|---|---|---|---|---|
| Xiong et al., 2011 [22] | Longitudinal | HRV, HA1c | IS | 34/71.7 ± 8.7 23M/11F | - | Hypertension (28) Diabetes (17) Hyperlipidemia (18) Ischemic heart disease (18) Smoking (9) Alcohol (7) | 14 | 2 months | Within 7 days after onset | NIHSS (on admission) mRS (on admission) Barthel index | In individuals with acute ischemic stroke, relatively substantial autonomic dysfunction is associated with a poor functional prognosis. |
| Chang et al., 2019 [64] | Longitudinal | HRVBF | IS | Patients; 19/67.6 ± 11.4 10M/9F Control; 16/67.2 ± 7.6 8M/8F | Beta-blockers (P. 2 /C. 3) Calcium channel blockers (P. 1/C. 1) | Hypertension (P. 13/C. 8) Diabetes (P. 9/C. 8) Heart disease (P. 7/C. 5) | - | 3 months | On admission | NIHSS (on admission) | HRVBF is a potentially effective treatment for AIS patients' autonomic dysfunction, cognitive decline, and psychological suffering. |
| Tang et al., 2020 [65] | Longitudinal | BPV, HRV, BRS | IS | 142/63.9 ± 10.2 125M/17F | Antihypertensives (93) Antiplatelet (139) ARB (43) ACEI (17) Beta-blockers (20) Calcium channel blockers (36) Diuretics (4) Statin (132) | Hypertension (101) Diabetes mellitus (51) Hyperlipidemia (61) Smoking (79) Alcohol (37) | 38 | 3 months | Within 7 days of an ischemic stroke | NIHSS (on admission) mRS (on admission and at 3 months) | In addition to the well-established predictive variables such as the National Institutes of Health Stroke Scale, a decreased low/high-frequency ratio of systolic BPV and impaired baroreflex sensitivity indicated an unfavorable stroke outcome. |

<div align="center">**Table 1.** *Cont.*</div>

| Authors, Year of Publication | Type of Study | Biomarker | Type of Stroke | Number of Participants/Mean Age (y)/Gender (M/F) | Medication (*n*) | Comorbidities/ Risk Factors (*n*) | Previous Stroke | Follow-Up Time | Time of Assessment | Scale of Stroke Severity and Prognosis/Clinical Outcome | Main Results |
|---|---|---|---|---|---|---|---|---|---|---|---|
| Constant-inescu et al., 2017 [66] | Longitudinal | HRV | IS | Right MCA IS 15/59.7 ± 10.3 8M/7F Left MCA IS 15/59.4 ± 8.43 7M/8F | - | Hypertension (25) | - | 3 months | Within 6 months post-stroke | mRS (3 months after stroke onset) | In stroke patients, the autonomic nervous system is predisposed to asymmetric, lateralized responses to various stimulation autonomic tests. In right-handed patients, right hemisphere stroke has a more marked sympathetic control on the HR than left hemisphere. |
| **Long-term Stroke Outcome (<1 Year)** | | | | | | | | | | | |
| Zhao et al., 2020 [24] | Longitudinal | HRV | AIS | 186/60 (53–66) 150M/36F | Calcium channel blocker (34) ACEI (9) Beta-blockers (3) | Coronary artery disease (29) Hypertension (108) Hyperlipidemia (25) Type 2 diabetes mellitus (55) Smoking (127) Alcohol (121) | 36 | 1 year | Within 1-week post ictus | NIHSS (on admission and at discharge) mRS (on admission, at discharge, and after 3, 6, 12 months) | AIS infarction basin, TOAST subtypes, and neurological outcomes at discharge are related to HRV evaluated after admission, suggesting a potential role for HRV in assessing AIS and identifying high-risk patients. |
| He et al., 2018 [42] | Longitudinal | HRV | AIS | 516/66.14 ± 10.11 253M/263F | Antihypert-ensives (321) Antiplatelets (126) Lipid-lowering medications (223) | Hypertension (367) Diabetes (154) Hyperlipidemia (274) Smoking (149) Alcohol (153) Family history of stroke (101) | - | 1 year | Within 24 h from onset | NIHSS (at baseline) | Decreased FD and END and 1-year RIS following an acute ischemic stroke are positively correlated. |

Table 1. *Cont.*

| Authors, Year of Publication | Type of Study | Biomarker | Type of Stroke | Number of Participants/Mean Age (y)/Gender (M/F) | Medication (*n*) | Comorbidities/ Risk Factors (*n*) | Previous Stroke | Follow-Up Time | Time of Assessment | Scale of Stroke Severity and Prognosis/Clinical Outcome | Main Results |
|---|---|---|---|---|---|---|---|---|---|---|---|
| Xiong et al., 2019 [67] | Longitudinal | HR, SBP, DBP | IS | 150/66.4 ± 9.9 106M/44F | - | Hypertension (93) Diabetes mellitus (51) Hyperlipidemia (65) Ischemic heart disease 15) Smoking (64) Alcohol (34) | 34 | Until discharge, 3 months and 1 year | 2–4 weeks post-stroke | NIHSS (on admission) mRS (3 months after stroke onset) | Significant autonomic dysfunction in patients with acute ischemic stroke is associated with a poor functional prognosis. |
| Lin et al., 2018 [68] | Longitudinal | BRS | AIS | 176/62.9 ± 12.3 135M/41F | - | Hypertension (149) Diabetes (74) Family history of stroke (63) Smoking (87) Alcohol (34) Hypercholesterolemia (141) Hypertriglyceridemia (84) | - | 12 months after stroke | Within 1 week after stroke | NIHSS (at 1 and 2 weeks after stroke and at discharge) mRS (1, 3, 6, and 12 months after stroke) | BRS within 1 week of stroke is a reliable indicator of complications during hospitalization for AIS and functional outcome (dependency) at 1 month after stroke. |
| Nayani et al., 2016 [13] | Cross-sectional | HRV | IS | 101/63 73M/28F | - | Systemic hypertension (65) Type 2 DM (31) Dyslipidemia (9) Coronary artery disease (24) Smoking (40) | - | 1 year | Within 2 weeks to 4 weeks post ictus | NIHSS (on admission) mRS (at discharge) | Greater frequency of autonomic dysfunction after stroke is linked to insular involvement and higher impairment at onset. Independent of the severity of the initial stroke, those with AD had a greater risk of infarct expansion, hospitalized cardiovascular complications, and a worse prognosis at 1 year. result |

<p align="center">**Table 1.** *Cont.*</p>

| Authors, Year of Publication | Type of Study | Biomarker | Type of Stroke | Number of Participants/Mean Age (y)/Gender (M/F) | Medication (*n*) | Comorbidities/ Risk Factors (*n*) | Previous Stroke | Follow-Up Time | Time of Assessment | Scale of Stroke Severity and Prognosis/Clinical Outcome | Main Results |
|---|---|---|---|---|---|---|---|---|---|---|---|
| Wang et al., 2017 [69] | Longitudinal | BPV | AIS | 873/NM 445M/428F | - | Hypertension (765) Diabetes mellitus (193) Hyperlipidemia (489) SCS (138) Coronary artery disease (116) Congestive heart failure (53) Chronic kidney disease (45) | 186 | 12 months | Within 7 days of onset | NIHSS (at onset and at 3 and 12 months) mRS (at 3 and 12 months) | High systolic or diastolic blood pressure within 7 days of the beginning of acute ischemic stroke was linked to neurological function recovery at 3 months, as well as recurrent stroke and composite cardiovascular events within 12 months. |
| **Chronic Stroke (>1 Year) Outcome** | | | | | | | | | | | |
| Sandset et al., 2014 [70] | Longitudinal | HR | IS, TIA | 3014/67.3 ± 8.3 1857M/1157F | Beta-blockers (815) Diuretics (1135) Valsartan (1513) | Diabetes mellitus (657) Smoking (572) Left ventricular hypertrophy with strain (535) History of atrial fibrillation (107) | 3014 | 4.5 years | Within the last 120 days of onset | - | The best predictor of recurrent stroke in high-risk, hypertensive patients with prior stroke or TIA was resting heart rate. |
| Leonarduzzi et al., 2018 [71] | Cross-sectional | HRV | IS | 173 | - | Atrial fibrillation (173) | | 47 ± 35 months | On admission | CHA2DS2-VASc | It was discovered that scattering coefficients were highly important in predicting IS, especially for patients not receiving antithrombotic therapy. |

**Table 1.** *Cont.*

| Authors, Year of Publication | Type of Study | Biomarker | Type of Stroke | Number of Participants/Mean Age (y)/Gender (M/F) | Medication (*n*) | Comorbidities/ Risk Factors (*n*) | Previous Stroke | Follow-Up Time | Time of Assessment | Scale of Stroke Severity and Prognosis/Clinical Outcome | Main Results |
|---|---|---|---|---|---|---|---|---|---|---|---|
| Verma et al., 2019 [72] | Longitudinal | PRV, HRV | IS | Patients; 41/64.4 ± 1.3 20M/21F Control; 29/65 ± 2 14M/15F | Antihypertensive medication (5-days prior to the study) | - | NM | - | On admission | NHISS (at the time of recording) mRS (at the time of recording) | Consideration of PRV as a substitute for HRV for assessing autonomic cardiovascular regulation when standing in stroke survivors should be performed with caution. |
| Webb et al., 2017 [73] | Longitudinal | BPV | IS | 472/66.2 ± 13.2 273M/199F | - | Diabetes mellitus (49) Family history (110) Hyperlipidemia (128) Smoking (73) | NM | 2 to 5 years | within 6 weeks of transient ischemic attack or non-disabling stroke | - | Beat-to-beat Independent of mean systolic blood pressure and risk variables, BPV predicted recurrent stroke and cardiovascular events, although short-term BPV on ambulatory blood pressure monitoring did not. |
| Nakanishi et al., 2017 [74] | Longitudinal | RHR, BP | IS | 2060/60 ± 11 1660M/400F | ACEI or ARB (2031) Calcium channel blocker (172) Diuretics (1683) Statins (1265) | Hypertension (1239) Diabetes (657) Smoking (363) Alcohol >2 oz/day (512) History of myocardial infarction (816) History of atrial fibrillation (76) | 252 | Up to 6 years (mean 3.5 ± 1.8 years) | Post-treatment | - | Those who have systolic heart failure in sinus rhythm may be more at risk for ischemic stroke if their RHR is lower. |

**Table 1.** *Cont.*

| Authors, Year of Publication | Type of Study | Biomarker | Type of Stroke | Number of Participants/Mean Age (y)/Gender (M/F) | Medication (*n*) | Comorbidities/ Risk Factors (*n*) | Previous Stroke | Follow-Up Time | Time of Assessment | Scale of Stroke Severity and Prognosis/Clinical Outcome | Main Results |
|---|---|---|---|---|---|---|---|---|---|---|---|
| Tang et al., 2012 [75] | Cross-sectional | BP | IS, HS, LS, US | 49/66.1 ± 7.0 29M/20F | - | - | NM | Until discharge | 4.5 ± 3.1 years post-stroke | NIHSS (on admission) Chedoke-McMaster Stroke Assessment | To determine those who have had a stroke with OH, a quick orthostatic tolerance test can be performed at the bedside. |
| Bodapati et al., 2017 [76] | Longitudinal | HRV | IS, HS | 884/75.3 ± 4.6 338M/546F | - | Left ventricular hypertrophy (45) Diabetes mellitus (149) History of heart disease (175) | - | ≤8 years | On admission | CHS-SCORE (on admission) | Information from 24-h Holter monitoring is significantly linked to the occurrence of stroke in older people living in the community who have a normal sinus rhythm. |
| **Depression due to AIS** | | | | | | | | | | | |
| Tessier et al., 2017 [43] | Longitudinal | HR, HRV | IS | 56/51.7 ± 13.0 39M/17F | - | Hypertension (23) Dyslipidemia (20) Diabetes (2) | - | 3 months | within the first week after stroke | NIHSS (on admission) Lesion volume mRS (after 3 months) | Patients who are unable to complete questionnaires may still be able to use resting HR measures collected in the early poststroke phase to aid in the early prediction of PSD and PSCI. |
| He et al., 2020 [42] | Longitudinal | HRV | IS | 503/65.93 ± 10.19 246M/257F | Antiplatelets (122) Antihypertensive (272) Lipid-lowering medications (195) | Hypertension (323) Diabetes mellitus (150) Hyperlipidemia (250) Smoking (138) Alcohol (149) Family history of stroke (101) | NM | 3 months | Within 72 h from symptom onset | NIHSS (on admission) | Due to the substantial correlation between decreased HRV and PSD, which is frequent following mild to moderate AIS, FD may be used as a reliable technique to predict PSD rhythms. |

**Table 1.** *Cont.*

| Authors, Year of Publication | Type of Study | Biomarker | Type of Stroke | Number of Participants/Mean Age (y)/Gender (M/F) | Medication (*n*) | Comorbidities/Risk Factors (*n*) | Previous Stroke | Follow-Up Time | Time of Assessment | Scale of Stroke Severity and Prognosis/Clinical Outcome | Main Results |
|---|---|---|---|---|---|---|---|---|---|---|---|
| Idiaquez et al., 2015 [77] | Longitudinal | BP, SCOPA-AUT | IS | Patients: 45/65.8 ± 11.7 28M/17F Control: 55/65.2 ± 15.1 19M/36F | Beta-blockers (C. 7/P. 6) ACEI (C. 17/P. 16) CCB (C. 7/P. 6) Diuretic (C. 6/P. 7) Antidepressant (C. 3/P. 5) | Hyperlipidemia (C.13/P. 9) Diabetes mellitus (C. 8/P. 14) Depression (C. 7/P. 16) Smoking (C. 14/P. 23) | - | Until discharge | Between 3 and 12 months after the first symptomatic stroke | NIHSS (on admission) OCSP | In hypertensive patients with mild ischemic stroke, autonomic symptoms, particularly gastrointestinal, urinary, and cardiovascular function, was much worse. |
| **Immunosuppression due to AIS** | | | | | | | | | | | |
| Brämer et al., 2014 [37] | Longitudinal | HRV, WBC, CRP, PCT, copeptin | IS | 240 | - | Infection (240) | - | 3 months | 3–5 days after onset | NIHSS (on admission) mRS (at 3 months) Barthel index | Infections in the later sub-acute period can be predicted using HRV indicators collected during the acute post-stroke period. The role of ANS activity in relation to immunomodulation and the emergence of infections after stroke is represented by HRV indices. |
| Brämer et al., 2019 [78] | Longitudinal | HRV | IS | 250/69 (60–78) 132M/118F | Beta-blockers (107) Antihypertensive drugs (156) | Coronary heart disease (42) Diabetes (58) Arterial hypertension (168) | NM | 3 and 5 days after stroke | 24 h after onset | NIHSS (on admission) | Infectious complications in the immediate post-stroke period are predicted by VLF. |
| Yuan et al., 2019 [36] | Longitudinal | ACh, AChE, ChAT | IS | 994/67.88 572M/422F | - | Hypertension (309) Diabetes mellitus (262) Smoking (233) Alcohol (233) | NM | 1 year | <24 h after onset | NIHSS (at baseline) mRS (at baseline) | In individuals with acute stroke, lymphocytes produced more ACh, and pneumonia was a probable outcome. |

### 3.3. Stroke Patient Groups and Demographic Profile

The total number of stroke patients included in all studies ranged from *n* = 12 to *n* = 3447 [45]. Across the 45 studies, 21 studies had a disease sample size between 1 and 100 patients, 7 studies between 101 and 200, 3 studies between 201 and 300, 0 studies between 301 and 400, 3 studies between 401 and 500, and 8 studies had a disease sample size larger than 500 patients. The mean/median patient age ranged from 51.7 years to 75.3 years.

### 3.4. Reference Groups

Across the 45 studies, the stroke patients were contrasted to demographically matched healthy individuals in only 10 studies, with the rest of them (35/45 studies) not including a healthy control group. None of the studies included a disease-control group other than stroke patients.

### 3.5. Time of Assessment

In 12 studies, the assessment was performed on admission, in 6 within 24 h of symptom onset, in 1 within 48 h of symptom onset, in 2 it was executed within 3 days from symptom onset, in 1 study within 5 days, in 10 studies within 7 days of symptom onset, in 1 study in the first 14 days, in 1 within the first 3 months, in 1 within the first 6 months, and in 12 studies assessment took place later.

### 3.6. Scales of Stroke Severity and Prognosis/Clinical Outcome

The NIHSS and modified ranking scale (mRS) were used simultaneously in 14 studies, NIHSS alone was used in 12 studies, and mRS was utilized exclusively in 2 studies. In the rest of the studies, a combination of scales of stroke severity and clinical outcome was used. More specifically, in one study, NIHSS was combined with Barthel Index (BI); in 3 it was combined with mRS and Barthel Index (BI). Moreover, in one study, NIHSS was combined with the Chedoke-McMaster Stroke Assessment, in another with Oxfordshire Community Stroke Project (OCSP) classification, and in another with the DSM-5 criteria. The following other scales used in one study each: CHS-SCORE, CHA2DS2-VASc, Functional Independence Measure, Rancho Los Amigos Scale, Fugl-Meyer Lower Limb Scale, Mini-Mental State Examination.

## 4. Discussion

A literature review was conducted over the last decade to assess the prognostic value of HRV (and other biomarkers related to AD) after stroke. Forty-five full-text original articles addressing the potential utility of evaluating HRV values for stroke prognosis were identified and grouped based on follow-up time and stroke subtype under investigation.

### 4.1. Early Neurological Deterioration (within 1st Week)

With respect to the effect of BP and HRV on neurological deterioration, Shimada et al. [31] concluded that ND after 48 h was related to greater changes in BP and HR and that an increased HR was associated with increased in-hospital mortality in ischemic stroke patients. In addition, it has been demonstrated that a poor prognosis is connected with sBP < 100 mmHg within 24 h of admission, with a drop in sBP > 26 mmHg occurring primarily in patients admitted to the hospital at night, which may be due to microcirculatory disturbance in the ischemic penumbra exposed to misery perfusion exacerbated by the greater BPV. Additionally, in patients with atherothrombotic cerebral infarction, occlusive vessels in the non-responsible sites were found more frequently in the patients with ND. They suggest that this may indicate that changes in the systemic circulation may affect cerebral circulation and cause ND when vasomotor reactivity is impaired because of arteriosclerosis. Similarly, Han et al. [45] investigated the connection between AIS patients' in-hospital mortality and their resting heart rate upon admission. The cumulative incidence rate of all-cause mortality was higher in patients with a higher resting heart rate (bpm $\geq$ 76),

as was the NHISS score. They discovered that in AIS patients, a 1.63-fold increased risk of in-hospital mortality is associated with a higher resting heart rate at admission. In addition, the AF status appeared to have a different impact on the prognostic value of resting heart rate in ischemic stroke patients. In patients without AF, patients with a heart rate of fewer than 76 beats per minute had a 2.39-fold higher risk of dying in the hospital than patients with a heart rate of fewer than 76 beats per minute. In AIS patients with AF, however, there was no independent correlation found between heart rate and in-hospital mortality. Using the heart rate as a continuous variable, sensitivity analyses confirmed similar outcomes. In contrast to patients without AF who had an ischemic stroke, patients with AF did not have a significant relationship between their heart rate and their mortality while in the hospital.

On another note, when characterizing HRV values in a heterogeneous group of severely injured patients with ABI, including stroke, Visitisen et al. [46] found that HRV was significantly reduced compared with the healthy control group. In addition, they observed a circadian pattern (6 pm–6 am) in SDNN and LF during the resting phase of the recording. They also examined time since injury (TSI) and systolic BP, and found a small but statistically significant association caused by the patients with the highest time since injury (89–190 days), although HRV, diastolic BP, and respiration were not related to TSI.

Considering the influence of the infarction type, Chen et al. [47] categorized stroke patients into the following two categories: Acute lacunar infarction (LAC) and acute atherosclerotic infarction of the great arteries (LAA) patients. They discovered that the LAA group had less parasympathetic activity and more sympathetic activity than the LAC group and that the LAA group was more likely to have a worse early outcome because of the lower parasympathetic activity. Autonomic modulations may play a significant role in outcomes in LAA patients, as the findings indicate that autonomic modulations behave differently in the LAA and LAC groups. The findings emphasize the link between stroke outcome and parasympathetic activity in various stroke causes. The relationship between autonomic function and stroke outcome has been linked by several different theories. The homeostasis of the visceral apparatus is influenced by parasympathetic modulation, which also regulates proinflammatory cytokines that are linked to early neurological deterioration (AIS). An animal model has shown that decreased parasympathetic modulation can also result in cerebral ischemia and a reduction in regional cerebral blood flow. As a result of decreased parasympathetic activity, patients with LAA who have cerebral edema and arterial stenosis may have marked cerebral hypoperfusion and a higher risk of cerebral ischemia. Future research needs to confirm the causal connection between parasympathetic modulation and cerebral perfusion.

On the contrary, Tang et al. [65] found that a reduced LF/HF ratio of systolic BPV was an independent predictor of poor outcome, suggesting a shift toward parasympathetic dominance. In addition, impaired BRS and reduced BPV were associated with poor stroke prognosis independent of mean arterial pressure. Regarding BP changes, Huang et al. [48] demonstrated that blood pressure response during orthostasis varies significantly among hospitalized AIS patients with stable medical conditions. This indicated that orthostatic blood pressure changes should be evaluated frequently and routinely in hospitalized patients and that a single measurement may underestimate the value. When screening for OH, patients with AIS, especially those who rank higher on the Fazekas scale and extracranial carotid stenosis, should have multiple measurements. Moreover, Usui H. and Nishida Y. et al. [49], supporting an earlier hypothesis, found a strong correlation between PA and the VLF part of HRV, especially with the nighttime VLF. They provide evidence that nighttime measurements are more reliable than 24-h measurements because stress, exercise, diet, and other factors may influence the results. In particular, high PA has been associated with higher activity ANS, which contributes to improved cardiovascular prognosis and overall health in stroke patients. It is therefore recommended that stroke patients increase their PA levels. No statistical difference was observed in the ECG or the blood pressure-derived heart period for either group, while stroke patients had a lower correlation between the two. However, the stroke group showed lower agreement between

the ECG and the blood pressure-derived SDNN. Despite the connection with VLF, there was no connection with the other components (LF-HF).

With respect to the prognostic significance of HRV as a way to measure autonomic function in AIS patients, Park et al. [50] evaluated its relationship to nutritional biomarkers. HRV parameters measured by 24 h ambulatory Holter electrocardiography were correlated with serum albumin, prealbumin, and transfer levels, which are nutritional biomarkers used to assess nutritional status. HRV parameters were lower in all frequency and time domains in the albumin-deficient and prealbumin-deficient groups than in the normal group. All frequency domain HRV parameters were lower in the transferrin-deficient group than in the normal group. In conclusion, according to HRV parameters, ANS dysfunction was correlated with poor nutritional status as measured by nutritional biomarkers such as serum albumin, prealbumin, and transferrin levels. The dysfunction of the ANS may indicate a link between stroke and inadequate nutrition. ANS dysfunction may be evidence of a correlation between poor nutritional status and stroke.

Apart from that, in their study, Huang et al. [52] used hemodialysis patients to examine the relationship between HRV and stroke both before and after dialysis. All predialysis HRV measurements in stroke patients were lower than those in controls, but they were not significantly different from those in patients without stroke. All postdialysis HRV measurements, except HF, significantly outperformed predialysis HRV values in stroke patients, whereas no postdialysis HRV increase was observed in stroke patients. In patients who did not have a stroke, changes in HRV were influenced by dialysis duration, serum iPTH, total cholesterol, and hs-CRP levels. Compared to healthy controls, hemodialysis patients had lower baseline HRV values. Albeit the genuine fundamental systems stay dubious, actuation of the renal afferents, underlying rebuilding of the heart and vasculature, as well as weakened reflex control of autonomic movement might assume significant parts in ESRD patients. In addition, their findings demonstrate a negative correlation between hypercholesterolemia and the LF/HF ratio in stroke-free patients.

The first study that used external counterpulsation (ECP) as a means of lowering AD after stroke was conducted by Xiong et al. [53] They found that HR increased after ECP in patients with AIS. ECP involves applying electrocardiographically triggered diastolic pressure to the lower extremities using air-filled cuffs. Although total power density (TP) increased in all patients, the LF R-R interval increased significantly after ECP only in patients with right infarction. The role of TP is not clear; however, the authors suggest that the increased LF RRI reflects improvement in parasympathetic and sympathetic NS regulation after ECP. The possible mechanism behind this is that ECP could, first, lead to vascular distention and stimulate baroceptors; second, improve cerebral perfusion, implying better recovery; third, reduce arterial stiffness, leading to increased vagal activity and BRS. Moreover, none of the patients had an insular lesion, suggesting that cardiovascular autonomic activation is triggered by extra-insular lesions that damage extra-insular central autonomic regions or their interconnecting fibers.

Additionally, to assess autonomic function, Xu et al. [54] proposed the use of PRSA, a new technique that can directly determine the acceleration capacity (AC) and deceleration capacity (DC) of the heart by analyzing the overall trend of the 24-h heart rate. It is believed that DC reflects vagal activity and AC reflects the sympathetic activity of the heart; hence, sympathetic and vagal activity can be detected simultaneously. In their study, they found that the patient group had lower SDNN, DC, and AC values, suggesting that sympathetic and vagal regulation were impaired in patients with hemispheric infarction. DC and AC were strongly correlated, suggesting that there was no significant change in sympathovagal balance in patients with stroke. Although the decrease in parasympathetic activity (DC) is consistent with previous studies, the decrease in sympathetic activity (AC) is a new finding that contradicts the existing literature. However, the R-R intervals of the AIS patients were shorter than those of the control subjects, which may indicate sympathetic hyperactivity. Given all this, they concluded that although both vagal and sympathetic modulation decrease after stroke, the combined effects of both lead to predominant sympathetic activity.

*4.2. Early Stroke Outcome (<1 Month)*

Concerning HRV as a prognostic biomarker, Chidambaram et al. [55] used frequency-domain measures (HF, LF, and LF/HF), and they discovered an increased LF/HF ratio of a significant portion of the stroke patients they examined, suggesting increased sympathetic activity. There was also a statistically significant decrease in survival rate with higher LF/HF ratio levels.

Furthermore, Tsai et al. [28] examined autonomic function in different subtypes of AIS and found that BRS function and HRV were significantly lower in those patients. Regarding BRS, the reduction was greater in large hemisphere and brainstem infarcts than in small, deep hemisphere infarcts, whereas there was no difference between left and right infarcts. Moreover, the LF/HF ratio was higher in stroke patients, suggesting sympathetic overactivity. In view of the above, they believe that BRS values are prognostic markers for 1-month outcome in stroke patients and recommend that patients be tested within 24 h of symptom onset. Similarly, in their cohort, Lai et al. [56] measured BRS, which was correlated with short-term outcome measures once during the acute stage of ICH (within three days). According to their findings, BRS is significantly lower in patients with poor outcomes than it is in sex- and age-matched controls, and it then rises after an acute spontaneous ICH event. BRS has previously been shown to be an independent predictor of long-term mortality following AIS. On Day 1, Day 4, and Day 10, the BRS of the poor outcome group was significantly lower than that of the gIn light of this finding, it is suggested that patients in subsequent studies of BRS or other cardiovascular autonomic changes in stroke patients should be tested within 24 h of the onset of symptoms to observe the most prominent differences. These differences may gradually diminish, suggesting that BRS blunting occurs transiently in ICH patients who are anticipated to have poor outcomes.

*4.3. Short-Term Stroke Outcome (<3 Months)*

In agreement with previously mentioned studies, A. Sethi et al.'s [23] findings suggested that HRV is positively associated with motor outcomes 3 months after stroke. The findings can be explained by the common neural circuit regulating the somatomotor and autonomic nervous systems. Specifically, they observed that HRV and initial upper extremity impairments were both equally significant variables to predict the movement of the affected extremity 3 months post-stroke. In contrast, HRV was more strongly associated with movement of the affected lower extremity 3 months after a stroke than initial impairment. They believe that is due to the fact that apart from the corticospinal pathways, lower extremity movement is also controlled by several nonspecific spinal interneurons common to the autonomic nervous system and lower extremity movement, hence the stronger relationship with HRV. Furthermore, HRV had a stronger association with 3-month motor outcomes than initial upper and lower extremity impairments in patients with severe initial motor impairments. Although the long-term motor outcome is difficult to predict, they concluded that HRV may be able to predict motor outcomes and assist doctors in effectively planning care to promote long-term independence.

Similarly, Zhang et al. [57] found that HRV of hospitalized stroke patients had a synchronous relationship with the recovery of motor function. The SDNN (r = 0.274; *p* = 0.032) and SDANN (r = 0.276; *p* = 0.031) values support the assessment and evaluation of the motor function of stroke patients in the chronic rehabilitation phase. In particular, the RR triangular index (r = 0.252; *p* = 0.05) in the time domain and the VLF domain correlated significantly with the change in motor functions. Regarding VLF, the correlation persisted from admission to approximately 3–4 months after stroke onset. Conversely, the RR triangular index of stroke patients was still associated with their motor functions after rehabilitation training. While an increase in the RR-triangle index could be relevant to the prognosis of motor function during long-term rehabilitation, a decrease in the RR-triangle index could be related to patient death during the acute phase. Compared with HRV parameters in the acute phase, HRV parameters in the rehabilitation phase have a stronger correlation with motor function. Because chronic stroke patients are in a stable

state, unlike acute stroke patients, the authors believe that the prognosis of motor outcomes is more reliable.

On the contrary, Graff et al. [58] suggested that continuous short-term recordings of HRV parameters, but not BPV, could be used to distinguish between groups with different neurological outcomes during the acute phase of ischemic stroke. Second, in AIS, a worse functional outcome is associated with a faster respiratory rate during the acute phase. For the 90-day outcome, neither nonlinear nor standard time domain parameters were of predictive value in our patients. In contrast, spectral HRV differentiated patients with poor and good outcomes. Moreover, there was no significant difference between the two groups in terms of blood pressure levels or blood pressure variability calculated as SD of SBP. Finally, their study demonstrated that respiratory rate during the acute phase was associated with functional outcomes in ischemic stroke. It is unclear which mechanisms underlie this connection. Dyspnea was reduced, pulmonary gas exchange was improved, and exercise tolerance was improved when the respiratory rate was slowed in congestive heart failure patients, according to previous research. Additionally, the fluctuations in left ventricle stroke volume caused by respiration-related changes in intrathoracic pressure led to increased variability in blood pressure at faster respiratory rates. In the case of a deficiency in buffering reflex mechanisms (such as decreased baroreflex function), the negative impact of a higher BP variability on neurological status may also be a possible explanation for patients with faster respiratory rates.

In contrast to that, regarding the clinical importance of HRV as a marker for patients with spontaneous ICH, Szabo et al. [59] found a decreased total volume of both HF and LF, demonstrating the general blunting of responses in the sympathetic-parasympathetic NS and baroreflex compartments. However, it still resulted in a low LF/HF and decreased HRV compared to healthy individuals, suggesting that LF is more significantly reduced than HF, resulting in a parasympathetic shift. This contradicts the assumption of previous studies, mainly investigating ischemic strokes, that stroke causes sympathetic dominance. They believe this is due to a phenomenon specific to ICH in the early phase, i.e., hemorrhage volume leading to a rapid increase in intracranial pressure, resulting in the observed parasympathetic dominance. This was in agreement with the findings of studies in patients with traumatic-severe brain injury with decreased LF/HF ratios. At the clinical level, for each unit of increase in normalized HF power, the poor outcome was an increase of 20%. In addition, for each unit of the LF/HF ratio, the outcome was expected to decrease by 93%. Considering all these factors, they argue that ANS can serve not only as a biomarker but also as a therapeutic target. On the same note, Tobaldini et al. [60] found that predominant vagal modulation was positively related to stroke severity in its very early stages, as shown by a higher 2 UV% in patients with NIHSS $\geq$ 14. Similar results apply to a worse midterm disability, with greater clinical impairment (mRS) corresponding to vagal modulation, consistent with previous studies. Specifically, with a higher 2 UV% and reduced 0 V% in the mRS 3–6 group, three months after stroke. In addition, they emphasized that there was no difference between anterior-posterior and insular/noninsular strokes and that the vagal shift was greater in the right hemisphere than left hemisphere lesions, although they believe this needs to be demonstrated in a larger number of patients. Reperfusion therapy appeared to have no effects on autonomic modulation.

Moreover, according to Miwa et al. [61] findings, functional disability is correlated with an increased mean HR and HR-average rate variability within 24 h of ICH. Hematoma expansion at 24 h and functional 3-month outcomes were consistently linked to HR-ARV. The association between HR-ARV and a worse functional outcome remained significant, even though they discovered evidence for effect modification according to the treatment group. In contrast, functional outcomes were correlated with mean HR within 24 h of ICH but not hematoma expansion. Therefore, an important outcome determinant can be an increase in mean HR and HR-ARV within 24 h of ICH. In the hyperacute phase of ICH, it would be crucial to not only rapidly reduce high HR but also to ensure that HR control is smooth and sustained. However, it is still unknown if HR modulation could improve

ICH prognosis. Their findings need to be confirmed, as well as the clinical implications and best HR management practices for the immediate post-ICH period. Additionally, Jeong et al. [62] found that in patients with AIS, increased tachycardia burden during the SU stay was linked to overall mortality and disability. Poor functional outcome was seen in approximately 68% of the patients in the highest quartile of tachycardia burden (6.0%). However, this study examined data on heart rate with a resolution of one minute. As a result, they could use a high-resolution data acquisition system to measure the burden of tachycardia in terms of the percentage of time tachycardia lasted per total monitoring time from continuous ECG monitoring. This study provides more precise data on the connection between tachycardia and clinical outcome thanks to its strengths. After controlling for potential factors that were associated with poor outcomes, their finding that tachycardia was strongly associated with poor functional outcomes suggests that tachycardia is at least a surrogate marker for multiple neurologic and/or medical worsening. Pathologic sympathetic activation during the acute phase of a stroke may be a potential predictor of subsequent cardiovascular and cerebrovascular events, which is one possibility. With coronary artery disease, an elevated heart rate has been linked to plaque rupture and recurrent myocardial infarction.

Additionally, Xiong et al.'s [63] study had similar findings using the Ewing battery test instead of HRV. In agreement with previously mentioned studies, they found that AIS led to AD; specifically, 76% of patients were diagnosed with severe AD. Moreover, patients with atypical, definite, or severe autonomic dysfunction were more likely to show poor improvement 2 after AIS on the BI scale (measuring performance on ADL). The mRS scale was probably not sensitive enough to detect clinically important differences in one of the studies that included only patients with mild stroke, leading experts to conclude that the BI scale may be a more reliable measure for assessing poststroke impairment. Another study by Xiong et al. [22] showed the same result 3 months after stroke. They believe that AD is due to cerebral hypoperfusion resulting from increased BPV.

On another note, Chang et al. [64] showed that HRV biofeedback training is feasible in patients with AIS. The finding that daily bedside HRVBF training sessions for four consecutive days (20 min/day) were effective in improving autonomic function, cognitive impairment, and psychological distress in participants is both surprising and exciting. Contemporary theoretical postulates and research findings suggest that HRVBF instructs participants to breathe in a manner (6 breaths/min) that promotes the rise and fall of heart rate in the same pattern as tidal breathing and improves parasympathetic function via vagus nerve stimulation and baroreflex reinforcement. Accordingly, HRVBF can eliminate the phase differences between heart rate fluctuations and respiration to strengthen the modulatory processes of the respiratory, cardiovascular, and emotional systems.

Finally, Tang et al., using a simple bedside orthostatic tolerance test, showed that 15% of community-dwelling stroke patients had OH, and this percentage increased to 20% when participants with symptomatic hypoperfusion were added. Compared with the 52% found in patients undergoing inpatient stroke rehabilitation, this represents a lower incidence of OH (but is consistent with other studies). This may be because patients in institutional settings are less mobile and, therefore, more susceptible to OH. Because of its prevalence, they believe that OH is clinically important after stroke because it contributes to stroke-related mobility limitations, balance problems, and fall risk in this already vulnerable group, especially since OH is associated with adverse cardiovascular (greater dyslipidemia and body weight) and cerebrovascular outcomes. However, most of them did not exhibit hypotensive symptoms, underscoring the importance of objective measurement of BP.

### 4.4. Long-Term Stroke Outcome (<1 Year)

The goal of Constantinescu et al.'s [66] study was to highlight the distinct influence of central control on autonomic HR modulation in patients with right and left medial cerebral artery ischemic stroke, using time and frequency domain parameters. In their analysis, they took into account time domain parameters such as RMSSD and pNN50 that

are linked to parasympathetic activity. They also took into account frequency domain parameters such as HF or LF, which have a more complicated origin that is still up for debate. However, they did not base their conclusion on the values of the LF and LF/HF ratios. Since nonlinear parameters are not typically used to define sympathovagal balance, their involvement is still up for debate. To highlight the sympathetic overactivation that is associated with reduced variability of the heart rate in a group of patients who are more likely to develop cardiac complications, they looked for a correlation between the linear parameters and the non-linear parameters of the HRV (such as SD1, which reflects the short-term variability, and DFA a1. Linear parameters (RMSSD, pNN50, SDNN, HFnu, LFnu, and LF/HF ratio) reflected cortical lateralization in the patients in their study as follows: parasympathetic dominance on the control of the HR in ischemic stroke patients with left MCA and sympathetic activation on the control of the HR in right hemisphere stroke patients. SD1s strong correlations with parasympathetic time domain parameters, both in resting state and during activation tests, suggested that it could be used as a reliable parameter for parasympathetic modulation in short-term (5 min) ECG recordings. They also found that the sympathovagal balance was positively correlated with the DFA a1 nonlinear parameter, which had a negative correlation with SD1 and HFnu and a positive correlation with LFnu and the LF/HF ratio. They did not find predictable outcomes with respect to ApEn and SampEn nonlinear boundaries during resting state and enactment tests.

Regarding the relationship between IS locations, etiology, prognosis, and AD, Zhao et al. [24] observed that most patients had large artery atherosclerotic infarction (LAA) in the left or right internal carotid artery (L or R ICA) or in the vertebrobasilar system (VB). They demonstrated that the LAA group had lower SDNN, RMSSD, and pNN50 scores than the non-LAA group did and that these scores were negatively correlated with clinical scores (mRS-NIHSS) at discharge. In addition, RMSSD and pNN50 scores were lower in patients with infarction of the VB arterial system than in patients with R-ICA infarction. The authors believe that this may be explained by a possible brainstem infarction in this area because previous studies have shown that brainstem infarctions more often lead to sympathetic shifts than hemispheric infarctions. Apart from this, in their study, SDNN, represented by the total level of ANS, RMSSD, and pNN50, which reflect parasympathetic activity, was found to be significantly decreased at discharge in patients with severe impairment as opposed to patients with mild impairment. The authors conclude that HRV parameters are related to the infarct basin, TOAST subtypes, and mRS score and could be used to assess and identify high-risk patients. On the same note, Nayani et al. [13], in their research on the relationship between stroke characteristics and the severity of autonomic dysfunction, suggested that the presence of AD was a predictor of in-hospital neurological deterioration and vascular complications with a less favorable outcome at discharge. This pattern was also maintained at a one-year follow-up, independent of the severity of the onset of the stroke. Neither the stroke subtype (LAA versus lacunar) nor the location had any effect on autonomic dysfunction in their stroke subjects, as has also been reported by other authors. They were unable to discover a relationship between stroke etiology and outcome, independent of stroke severity, due to the small numbers in each subtype. In the majority of series, lacunar strokes have been linked to excellent outcomes. However, their study was underpowered to see regardless of whether more serious strokes of lacunar etiology had AD.

Apart from that, using HRV by fractal dimension (FD), He et al. [67] demonstrated a positive correlation between decreased FD, early neurological deterioration (END), and a one-year RIS following an acute ischemic stroke. Given the increased risk of recurrent stroke, their findings demonstrated the significance of an early diagnosis of autonomic dysfunction in stroke survivors. FD might have possible prescient worth in the risk definition of ischemic stroke. ECG monitoring should be performed regularly for stroke unit admissions, and FD analysis only requires about 15 min worth of ECG data. This suggests that the management of acute strokes may benefit from this strategy. Moreover, regarding the assessment of autonomic function, Xiong et al. [68] suggest in another study that the Ewing battery test is

a reliable and potentially more accurate method for determining the severity of AD, even in chronic stroke.

When assessing BRS as a biomarker for the AS, Lin et al. [33] investigated its significance in predicting patient mortality and subsequent events up to a year after an AIS. BRS is a crucial regulator of arterial pressure and a first-of-its-kind independent predictor of short-term functional outcome (modified Rankin scale at one month after stroke) and hospitalization complications (pneumonia, UTI). In their research, hypertension, coronary artery disease, myocardial infarction, chronic heart failure, and stroke outcomes have all been predicted by BRS. Similarly, Wang et al. [69] sought to examine the association between midterm (within 7 days of onset) BPV and clinical outcome after AIS. Consistent with previous studies, they found that midterm systolic and diastolic BPV was correlated with an increased risk of recurrent stroke and cardiovascular events during a 12-month follow-up period. In addition, they emphasized the importance of assessing neurological improvement 3 months after stroke. They found that higher midterm diastolic and systolic BPV correlated negatively with good neurological recovery 3 months after stroke and positively with neurological deterioration and poor functional outcome.

### 4.5. Chronic Stroke (>1 Year) Outcome

Regarding the role of resting HR as a predictor of stroke in high-risk patients with previous stroke or transient ischemic attack, Sandset et al. [70] suggested that the relationship between resting heart rate and recurrent stroke may be a marker of underlying and undetected cardiovascular disease or that it can represent sympathetic hyperactivity and/or parasympathetic withdrawal, ventricular dysfunction, or hypertension. Furthermore, despite the findings in previous studies suggesting that lowering HR with ivabradine reduced both cardiovascular events and cerebral infarct size, they found a numerical but statistically nonsignificant reduction of the stroke risk.

Moreover, by collecting information from 24-h Holter monitoring, Bodapati et al. [76] suggested that HRV was significantly associated with the occurrence of stroke in community-dwelling older adults with normal sinus rhythm, even after calculating their stroke risk using a validated clinical score and may improve risk stratification for stroke occurrence in this population. Also of note is the difference between HRV risk factors associated with stroke and HRV risk factors previously associated with cardiovascular mortality risk (CHS). Higher numbers of atrial and ventricular ectopy were also risk markers for worse cardiovascular outcomes (CHS), but no significant difference in the number of atrial or ventricular ectopy was found between those who suffered a stroke and those who did not. Similarly, the mean 24-h heart rate, another risk factor for adverse cardiovascular outcomes, did not differ between participants who had suffered a stroke and those who had not.

On the same note, Leonarduzzi et al. [71] suggested that features based on the scattering transform and multiscale entropy could be useful as predictors of ischemic stroke, especially in patients who are not receiving antithrombotic therapy. The results demonstrate the significant discriminatory power provided by a cascade of straightforward operators in the scattering transform and emphasize that relevant information is encoded in nonlinear dynamics that are not accessible through simpler spectral techniques. They also demonstrate that when S-SVM performs feature selection and classification simultaneously, its classification performance is enhanced. Especially noteworthy are the findings that activity in the HF range, which was previously thought to be random noise under AF, is discriminant.

Furthermore, utilizing the pulse rate variability (PRV) as a surrogate for HRV, Verma et al. [72] investigated the effects of AIS on the functional capacity of PRV to monitor autonomic cardiovascular control. The most important finding of their study was a decrease in the agreement between HRV and PRV during standing in stroke survivors, in time and frequency domains. Both the control group and the AIS group had decreased RR intervals in standing, while the stroke group had lower values than the non-stroke group in both standing and sitting. This is due to the readjustment of baroreceptors as a result of AIS and

possibly the decreased PA in stroke survivors. In contrast, they observed no differences in BP values while sitting; however, only the stroke group showed an increased BP while standing. The stroke survivors also had lower pulse arrival time values, underscoring the effects of stroke on vascular tone and blood flow. In addition, the stroke group showed a lower correlation between HRV and PRV, both in the frequency domain, especially in the LF/HF ratio, and in the time domain (pNN50), suggesting that PRV may not be reliable for measuring the function of ANS in the orthostatic challenge. Similarly, Webb et al. [73] observed that during 5-min beat-to-beat BP monitoring, BPV predicted the risk of recurrent stroke and all cardiovascular adverse effects, with a 4-fold increase in risk between the lowest and highest quartiles of the patients. In addition, the data show that antihypertensive treatment has a poor prognosis despite good control of the mean BP, although the benefits of some drugs in stroke prevention may be due in part to reduced variability in systolic BP. They suggest that a rapid, 5-min, beat-to-beat assessment of BPV may have similar prognostic power to the daily home measurement of BP and thus may be a useful index for guiding antihypertensive treatment.

On the same track, considering patients with heart failure with reduced left ventricular ejection fraction (HFrEF) in sinus rhythm who are treated with the current recommended medical regimen, which includes beta-blockers, Nakanishi et al. [74], suggested a higher risk of ischemic stroke in those patients when they show low RHR. A high RHR is associated with increased mortality and hospital re-admissions in patients with HFrEF who are in sinus rhythm, and beta-blockers significantly improve the outcome. Several meta-analyses have demonstrated a stronger effect on survival for RHR than the beta-blocker dose achieved, although the benefits of beta-blockers may not be entirely related to RHR reduction. Consequently, it has become a common clinical assumption that the RHR-lowering effect of beta-blockers is a prerequisite for the following beneficial effect: The benefit is greater the slower the RHR. However, given that low RHR is typically associated with lower rates of mortality and hospital admission in these patients, their finding was unexpected. In other clinical settings, where contradictory results have been reported, the connection between heart rate and stroke remains a mystery.

Finally, Tang et al. [75], using a simple bedside orthostatic tolerance test, showed that 15% of community-dwelling stroke patients had OH, and this percentage increased to 20% when participants with symptomatic hypoperfusion were added. Compared with the 52% found in patients undergoing inpatient stroke rehabilitation, this represents a lower incidence of OH (but is consistent with other studies). This may be because patients in institutional settings are less mobile and, therefore, more susceptible to OH. Because of its prevalence, they believe that OH is clinically important after stroke because it contributes to stroke-related mobility limitations, balance problems, and fall risk in this already vulnerable group, especially since OH is associated with adverse cardiovascular (greater dyslipidemia and body weight) and cerebrovascular outcomes. However, most of them did not exhibit hypotensive symptoms, underscoring the importance of objective measurement of BP.

*4.6. Depression Due to AIS*

To assess autonomic function, Tessier et al. [43] used time-domain (mHR, RMSSD) and frequency-domain (HF, LF, LF/HF) measures of resting HR and HRV and in correlation with previous studies showed that decreased HRV and sympathetic hyperactivity are related to higher severity of depressive symptoms, from the first week of stroke. Notably, RMSSD, reflecting the parasympathetic nervous system, is negatively associated with depressive scores, whereas high LF /HF in the early phase predicts higher levels of depressive symptoms at a 3-month follow-up. HRV measurements also correlate with cognitive functioning 3 months later. This highlights its importance as a biomarker, especially for patients with aphasia and cognitive disorders who cannot complete questionnaires, and that it could identify patients who will develop depression in the chronic phase. It was not clear whether the HR parameters measured in the first week after the stroke were exclusively related to the stroke. They are either due to direct damage to the corticolimbic

network or reflect the pre-stroke state, which makes it difficult for patients to overcome the effects of stroke. In conclusion, both studies demonstrated that when questionnaires were ineffective due to severe aphasia or cognitive impairment, HRV-FD measurements could help predict PSD.

Concerning the commonness of depression among stroke patients, He et al. [42] investigated the relationship between the fractal dimension (FD) at admission and the onset of depression seven days later and three months later in patients with acute mild-moderate ischemic stroke (PSD). The study demonstrated that a lower FD value was associated with the presence of PSD, as was a significantly increased risk of early-onset PSD status and 3-month PSD in patients. Stroke survivors frequently experience PSD, which can hinder recovery. In this study, FD was used to assess autonomic function and investigate the connection between FD and post-AIS depression. The groups of patients with PSD and those without PSD were significantly different, according to their findings. When compared to the group without PSD at 7 days, the depressive sample had lower FD values, was younger, and had a higher NIHSS. The severity of the stroke and lower FD value remained consistently associated with PSD at 3 months. Lower HRV may indicate greater susceptibility to PSD. In ischemic stroke, FD may have potential predictive value. Moreover, the main results of Idiaquez et al.'s [77] research were a correlation between depression and the total autonomic symptoms score and domain scores for gastrointestinal, urinary, and cardiovascular function in poststroke patients and patients without stroke. In addition, they discovered that the urinary symptoms score was linked to depression scores and that nocturia and urge incontinence were common in the post-stroke period. There is less information available regarding orthostatic intolerance symptoms in comparison to cardiovascular autonomic symptoms. There may be neurogenic and non-neurogenic causes of OH in poststroke patients. They concluded that hypertensive people who had a minor stroke had more autonomic symptoms than hypertensive people who did not have a stroke. Autonomic symptoms scores did not correlate with stroke severity, lateralization, or subtype. Additionally, there was a correlation between autonomic symptoms and depression.

### 4.7. Immunosuppression Due to AIS

Regarding the predictability of poststroke infection, Brämer et al. [37] found that HRV indices obtained in the acute phase after stroke were able to predict infections in the subsequent subacute phase. HRV indices represent the role of ANS activity in the context of immune modulation and the development of infections after stroke. They would be readily available in routine ECG monitoring systems for stroke. Their continued consideration would allow earlier identification and, thus, timely appropriate treatment of developing infections. Biomarkers obtained from poststroke blood samples, such as WBC, CRP, PCT, and copeptin, were all independent predictors of pneumonia, UTI, and other infections. The authors concluded that combining established inflammatory factors (WBC, CRP) with a biomarker for stress (copeptin) or a biomarker for bacterial infection (PCT) is likely to reflect the complexity of infection better than one biomarker alone and may lead to more accurate prediction of an incipient but not yet clinically apparent infection. Apart from that, Brämer et al. [78] performed their study using HRV within 48 h of stroke onset to predict stroke-related infection, SIRS, and severe sepsis in the post-acute period. Their goal was to identify patients at risk before clinical symptoms and paraclinical markers of infection become apparent to allow early and appropriate therapy. Their results showed that VLF predicted the development of infectious complications after stroke with similar accuracy to clinical risk factors at admission, such as stroke severity and insular cortex involvement. In their study, reduced VLF was best at predicting poststroke infections. This finding is consistent with the results reported regarding the development of infections and inflammation in different populations without stroke.

As far as stroke-induced activation of the parasympathetic cholinergic anti-inflammatory pathway and its association with poststroke mortality, Yuan et al. [35] demonstrated that

peripheral blood mononuclear cell (PBMC)-derived ACh was increased within the first 24 h after acute ischemic stroke. In addition, decreased expression and catalytic activity of acetylcholinesterase (AChE) were observed, suggesting an overabundance of stroke-induced cholinergic activity. Given this, AChE was suggested by the authors to be a marker of inflammation and a prognostic factor for rehabilitation. Furthermore, their study showed that stroke-induced pneumonia was associated with a significant increase in PBMC-derived ACh, emphasizing the role of cholinergic modulations as early mediators in the peripheral response to pulmonary infection.

## 5. Limitations

Our systematic review is not without limitations. First, it is possible that other studies published by the same research teams used data from the same cohorts, a possibility that we cannot completely rule out. Second, patients in most studies suffered a stroke of mild to moderate severity because patients who were severely debilitated or deemed unsuitable for an MRI and patients with dementia who were unable to give consent were usually excluded, limiting the generalizability of the results to the entire stroke population. Third, the influence of sociodemographic characteristics on the connection between HRV and patients' functional outcomes has not been thoroughly investigated. For this reason, we presented a detailed account of patients' sociodemographic and clinical data, which may provide an indirect estimate of their role in acute stroke and inform the design of future studies.

## 6. Conclusions

Considering all aspects, the present review provides an overview of the potential clinical applications of HRV (and other biomarkers related to AD) as a prognostic tool in acute stroke. Our 22 findings support the beneficial use of HRV measurement in poststroke prognosis and suggest that a biomarker-based approach using HRV assessment may provide important insights into the recovery potential of stroke survivors and greatly facilitate individualized stroke treatment. HRV appears to serve as a surrogate marker of stroke severity and reliably differentiates between patients with a good prognosis and those with an unfavorable functional outcome. Interestingly, HRV levels could independently predict both clinical outcome and mortality because it has been found that decreased HRV levels are usually associated with poor prognosis after stroke. It should also be stated that although some studies support the widely held belief that a high vagal and a low sympathetic tone correlate with a favorable course, several studies have found the opposite. This may reflect the complexity and target-specificity of autonomic regulation and may also relate to the fact that HRV only tells a certain extent about vagal and sympathetic cardiac tone and nothing else. It is of great importance that the above data are not limited to cases of acute ischemic stroke but that similar results have also been reported in patients with TIA and HS-ICH. Considering that HRV assessment can provide valuable predictive information beyond clinical variables, it could significantly improve the discriminative accuracy of widely used validated prognostic scores and thus optimize overall stroke management. Additional studies in stroke patients on the relationship between HRV and recovery probability are recommended to further elucidate this clinically important relationship.

**Author Contributions:** I.O. and E.M. reviewed the literature, screened the abstracts of the reference list, deleted duplicates and citations not meeting the inclusion criteria, and assessed the articles; K.V. solved any disagreements regarding screening or the selection process; E.P. and A.B. wrote the first manuscript; S.K. (Stella Karatzetzou), S.K. (Sofia Kitmeridou), A.S., A.G., A.T., C.K. and S.I. reviewed the tables, the presentation of the data, and the methodology. The corrected version was discussed collegially. K.A., F.C., N.A. and D.T. wrote the final version. All authors have read and agreed to the published version of the manuscript.

**Funding:** We acknowledge the support of this work by the project "Study of the interrelationships between neuroimaging, neurophysiological and biomechanical biomarkers in stroke rehabilitation (NEURO-BIO-MECH in stroke rehab)" (MIS 5047286), which was implemented under the action "Sup-

port for Regional Excellence" funded by the operational program "Competitiveness, Entrepreneurship and Innovation" (NSRFm2014-2020) and co-financed by Greece and the European Union (the European Regional Development Fund).

**Institutional Review Board Statement:** Not applicable.

**Informed Consent Statement:** Not applicable.

**Data Availability Statement:** All data discussed within this manuscript are available on PubMed.

**Conflicts of Interest:** The authors declare no conflict of interest.

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
