# Peer review of "Exploring the Utility of Autonomic Nervous System Evaluation for Stroke Prognosis"

_2035-8377, doi:10.3390/neurolint15020042_

Round 1

Reviewer 1 Report

This is a timely review helping to address the important issue of personalized medicine in the area of stroke prognosis, therapy and rehabilitation. The authors collected and analyzed a reasonable number of recent studies on the predictive validity of autonomic nervous system markers in stroke patients. Special emphasis was on the use of heart rate variability (HRV). Although overall well done, I have a few suggestions and minor points which need correction or clarification.

Major:

1) In the conclusion, HRV is highlighted as a proxy of autonomic balance.  However, in the Discussion several studies were analyzed which apparently are at odds with the widely held belief that a high vagal and a low sympathetic tone correlate with a favourable course (Tang et al., Xu et al., Szabo et al.). This may reflect the complexity and target-specificity of autonomic regulation (see Jänig 2022, The Integrative Action of the Autonomic Nervous System, Cambridge University Press) and may also relate to the fact that HRV only tells to a certain extent about vagal and sympathetic cardiac tone and nothing else. I suggest to take account of these inconsistencies by including a critical caveat.  

Minor:

2) Introduction, line 36: skip "old"

3) Introduction, line 77: renin

4) Introduction, line 111: "...located in gray matter..." all the structures named afterwards (hypothalamus, ventrolateral madulla, sensorimotor cortex) represent "gray matter"; thus it can be skipped

5) INtroduction, lines 166, 167: please avoid the term "parasympathetic afferents"; probably vagal afferents were meant

6) Discussion, lines 324, 325: which patients had a higher risk: those with heart rate greater or smaller than 76 BPM? 

Author Response

Dear Reviewer,

We are extremely grateful for your prompt response and the time spent reviewing our manuscript.

Appropriate modifications were made to the text as follows:

1) In the conclusion, HRV is highlighted as a proxy of autonomic balance.  However, in the Discussion several studies were analyzed which apparently are at odds with the widely held belief that a high vagal and a low sympathetic tone correlate with a favourable course (Tang et al., Xu et al., Szabo et al.). This may reflect the complexity and target-specificity of autonomic regulation (see Jänig 2022, The Integrative Action of the Autonomic Nervous System, Cambridge University Press) and may also relate to the fact that HRV only tells to a certain extent about vagal and sympathetic cardiac tone and nothing else. I suggest to take account of these inconsistencies by including a critical caveat.

Your valuable comment was added to our conclusion section

2) Introduction, line 36: skip "old"

Skipped

3) Introduction, line 77: renin

Corrected

4) Introduction, line 111: "...located in gray matter..." all the structures named afterwards (hypothalamus, ventrolateral madulla, sensorimotor cortex) represent "gray matter"; thus it can be skipped

Skipped

5) INtroduction, lines 166, 167: please avoid the term "parasympathetic afferents"; probably vagal afferents were meant

Corrected

6) Discussion, lines 324, 325: which patients had a higher risk: those with heart rate greater or smaller than 76 BPM?

Corrected

Looking forward to your follow up comments.

Yours Sincerely,

Dr Dimitrios Tsiptsios

Neurologist - Clinical Neurophysiologist

Democritus University of Thrace, Greece

Reviewer 2 Report

Using Medline and Scopus the authors did a literature search (articles in English) in the past decade on the relation between HRV and stroke prognosis.  They found sufficient evidence that in addition to HRV, autonomic dysfunction (AD) suffices to give a reasonable and reliable prediction of stroke prognosis. 

The methodology was reliable and vigorous, although in my opinion a more lengthened time of literature search, say in the past 30 years, is more desirable. The authors have mentioned a number of limitations in this study, especially about very severe stroke patients. The latter had therefore been excluded.

The authors also mentioned effects of sociodemographic factors on HRV-clinical outcomes, which are important criteria that the authors have been paying attention to in this article and in future studies. 

Author Response

Dear Reviewer,

We are extremely grateful for your prompt response and the time spent reviewing our manuscript.

Looking forward to working with you again in the near future

Yours Sincerely,

Dr Dimitrios Tsiptsios

Neurologist - Clinical Neurophysiologist

Democritus University of Thrace, Greece